Effect of underwater visual survey methodology on bias and precision of fish counts: a simulation approach

Pais Miguel Pessanha mppais@fc.ul.pt
Cabral Henrique N.
MARE—Marine and Environmental Sciences Centre, Faculdade de Ciências, Universidade de Lisboa , Portugal
Lino Pedro
Electronic publication date: 2018 Jul 30
Publication date: 2018
Volume: 6
Electronic Location ID: e5378
Received 2018 Feb 27; Accepted 2018 Jul 13
Copyright: ©2018 Pais and Cabral
Copyright year: 2018
Copyright holder: Pais and Cabral
License: This is an open access article distributed under the terms of the Creative Commons Attribution License, which permits unrestricted use, distribution, reproduction and adaptation in any medium and for any purpose provided that it is properly attributed. For attribution, the original author(s), title, publication source (PeerJ) and either DOI or URL of the article must be cited.
License URL: https://creativecommons.org/licenses/by/4.0/

Keywords: Sampling, Individual-based model, Fish behaviour, Reef fish, Underwater visual census, Agent-based model, Fishcensus, Computer simulation

Funding: Postdoctoral grant SFRH/BPD/94638/2013 Fundação para a Ciência e Tecnologia (FCT) Research funded through postdoctoral grant SFRH/BPD/94638/2013 attributed to MP Pais and MARE’s strategic project UID/MAR/04292/2013 both by Fundação para a Ciência e Tecnologia (FCT). The funders had no role in study design, data collection and analysis, decision to publish, or preparation of the manuscript.

==============================
Bias in underwater visual census has always been elusive. In fact, the choice of sampling method and the behavioural traits of fish are two of the most important factors affecting bias, but they are still treated separately, which leads to arbitrarily chosen sampling methods. FishCensus, a two-dimensional agent-based model with realistic fish movement, was used to simulate problematic behavioural traits in SCUBA diving visual census methods and understand how sampling methodology affects the precision and bias of counts. Using a fixed true density of 0.3 fish/m2 and a fixed visibility of 6 m, 10 counts were simulated for several combinations of parameters for transects (length, width, speed) and point counts (radius, rotation speed, time), generating trait-specific heatmaps for bias and precision. In general, point counts had higher bias and were less precise than transects. Fish attracted to divers led to the highest bias, while cryptic fish had the most accurate counts. For point counts, increasing survey time increased bias and variability, increasing radius reduced bias for most traits but increased bias in the case of fish that avoid divers. Rotation speed did not have a significant effect in general, but it increased bias for fish that avoid divers. Wider and longer transects and a faster swim speed are beneficial when sampling mobile species, but a narrower, shorter transect with a slow swim is beneficial for cryptic fish.

Introduction

Underwater visual census (UVC) methods are used worldwide to survey shallow aquatic habitats and are particularly known for supporting conservation and fisheries management decisions on temperate and coral reefs (Caldwell et al., 2016).

UVC methods can be generically classified into stationary point counts, transects and timed swims. In a stationary point, a diver stays in the same spot and surveys a fixed radius for a given time (Bohnsack & Bannerot, 1986). In the transect, the diver swims in a straight line at a constant speed for a given distance (or time) and counts organisms within a pre-determined transect width (Brock, 1954), or while estimating the distances to each organism (Thomas et al., 2010). On the timed swim, a diver swims along a random path, or changes direction at fixed intervals, counting organisms along the way (Jones & Thompson, 1978; Kimmel, 1985). Each of these techniques has advantages and disadvantages, and some might be more suited for a particular purpose or species (Lincoln Smith, 1989; Kulbicki et al., 2010). In addition, UVC methods have different configurations, such as different transect lengths and widths, observation ranges or survey times. This results in a vast amount of combinations that may not be directly comparable, making this choice a very important one for future re-use of survey data (Sale & Sharp, 1983; Cheal & Thompson, 1997; Kulbicki et al., 2010). However, criteria for choosing a UVC method are often independent of the research question, purpose or species of interest, and are deeply rooted in local tradition or institutional adoptions for long-term studies (Caldwell et al., 2016). This contrasts with the fact that UVC performance is deeply linked to species behavioural traits (Sale & Sharp, 1983; Kulbicki, 1998; MacNeil et al., 2008b; Bozec et al., 2011; Pais & Cabral, 2017). For a sampling method, performance results from the combination of precision (dispersion of estimates) and bias (inaccuracy of estimates). A method that has great precision should be favoured for hypothesis testing, while a method that leads to lower bias should be used in biomass estimates for fisheries management or conservation (Trebilco et al., 2011; Jones et al., 2015).

There is a long history of efforts to understand how sampling performance is affected by different species and methods in the field. Cryptic and biomimetic species tend to have low detectability and thus lead to underestimation (Willis, 2001; Bozec et al., 2011), while schooling species can be difficult to count accurately and can also lead to great variability in estimates depending on whether or not a school is found on a particular sample (Christensen & Winterbottom, 1981; MacNeil et al., 2008b; Kulbicki et al., 2010). Shyness and boldness towards observers is also a behavioural trait known to affect counts. The probability of recounting bold fish is increased, while shy fish are more likely to be missed, as they are often found farther away from the observer and therefore less detectable (Samoilys & Carlos, 2000; Bozec et al., 2011; Prato et al., 2017). This decay of detectability with distance is the foundation of the distance sampling method applied in visual census of many organisms (Thomas et al., 2010; Katsanevakis et al., 2012), but its application in UVC of fish is still uncommon (Caldwell et al., 2016). Reasons for this include the difficulty of estimating distances underwater, the proximity of the observer to the organisms being counted and visibility constraints, which sometimes lead to the need to establish a hard limit on the observation area that is so close to the centre that a constant detection function can be assumed (Bozec et al., 2011). While focusing on detectability, studies often ignore that UVC methods are usually non-instantaneous, which can lead to overestimated abundance of moving fish, as new fish enter the sample area after the observation started. This source of bias is known and acknowledged, but very difficult to measure in the field and has been mostly identified on computer simulations (Watson, Carlos & Samoilys, 1995; Ward-Paige, Flemming & Lotze, 2010; Glennie, Buckland & Thomas, 2015; Pais & Cabral, 2017). This effect has been shown to exceed the effects of behavioural traits in simulated transects and stationary points, except in the case of sedentary or cryptic species, which are slow moving or stationary (Pais & Cabral, 2017).

It has been repeatedly suggested that different species, or at least different behavioural traits, should be approached with different methods, and this has been applied in some cases (De Girolamo & Mazzoldi, 2001; Henriques et al., 2013a; Henriques et al., 2013b; Pais et al., 2013; Pais et al., 2014; Prato et al., 2017). However, choosing a method that minimises bias is not an easy task, particularly in the field, where the true density of a species is very hard or impossible to quantify. Often field studies that compare methods tend to assume one of them is at least more accurate (Sale & Sharp, 1983; Kimmel, 1985; St. John, Russ & Gladstone, 1990; Edgar, Barrett & Morton, 2004; Bennett et al., 2009), but it is difficult to ensure it is objectively accurate, since UVC methods are known to be biased and other techniques used as “controls” (capture-resight, fishing gears, poisoning, baited video, etc.) are also prone to errors (Caughley, 1974; Katsanevakis et al., 2012).

To identify a suitable method for a species, it is important to use a tool that can address bias due to behaviour, but also due to non-instantaneous sampling, and this is where simulation tools can be useful. In this study, the agent-based simulation model FishCensus (Pais & Cabral, 2017) is used to analyse the effect of various UVC sampling parameters (e.g., sampling unit dimensions, observer speed) on bias and precision, both for strip transects and stationary point counts, and identify the best solutions for fish with different behavioural traits.

Materials and Methods

Model overview

The FishCensus model simulates how different fish behaviours affect density estimates in common underwater visual census methods. A flexible vector-based fish movement algorithm can be adjusted to match behavioural patterns of species or groups.

This study used version 2.0 of the FishCensus model, programmed in NetLogo 6 (Wilensky, 1999). A full description following the ODD (Overview, Design concepts, Details) protocol for describing individual-based models (Grimm et al., 2010) is available in Pais & Cabral (2017), as supplemental material to this paper (Article S1) and in the COMSES repository (https://www.comses.net/codebases/5305/), where the latest version of the model, all source code and documentation are available.

The model is spatial, two-dimensional and has two types of moving agents, divers and fish. The model landscape is represented by a grid of squares with 1 m sides that have no variables directly affecting agents. Depth is ignored (assumed constant) and maximum underwater visibility was set to 6 metres and remained constant. This is a common, albeit low-end visibility for UVC, but it is still adequate for surveys and limits sampling unit sizes to a region where constant detectability with distance can be assumed (Bozec et al., 2011). The landscape size was set to 20 × 80 squares (1,600 m2) to allow for enough buffer space outside the sample area. Fish wrap around when they reach the edges to avoid artificial gathering near walls. There are two levels on the time scale. Fish and diver movements use a time step representing 1/10 of a second and all other procedures in the model are based on a time step of one second.

Fish movement is based on the sum of vectors that define “urges” (avoid diver, align with schoolmates, centre position in school, schoolmate spacing, wander, rest, cruise). The magnitude of the urge vectors can be weighted to rank them in terms of importance, and these weights are stored as fish attributes. Another important fish attribute for the movement model is a constant used to estimate friction drag that is calculated from fish size and creates a deceleration vector. A weighted sum of all the urge vectors at every time step determines the velocity vector on the next time step (Wilensky, 2005). A description of all vectors is available in the full description (Article S1). A set of weights for all vectors defines a behavioural state, and fish can have up to four states stored in a list, each with an associated probability (e.g., a “searching” state would give more weight to the urge to wander randomly). If the fish has more than one behavioural state on its repertoire, a new one is picked every 10 model seconds based on a weighted random pick with replacement, so a very frequent behaviour can be picked multiple times sequentially. A detectability parameter can also be set, establishing the probability of being visible to the diver. Whether a fish is hidden from the diver is decided after a behaviour change, for every fish independently, as a Bernoulli trial based on detectability.

At the start of the simulation, fish are randomly placed on the environment, with a random heading. Every fish picks the first behaviour on the repertoire and the movement submodel is run for 200 cycles (20 model seconds) before the diver is placed to stabilise initial fish locations and form schools when applicable. A diver performing a transect moves forward at a pre-determined speed, while during stationary point counts the diver rotates clockwise on a fixed point with a pre-defined angular speed, also every 1/10 of a second. Every second, the diver counts fish within the area delimited by view angle, sample limits and maximum visibility. The diver prioritises closest fish and counts with a saturation limit of 3 fish per second. Counted fish are memorised and are not recounted while they remain visible (can be recounted if they leave and re-enter the field of view). Once the diver has finished the sample, the model run is over, and density is calculated by dividing the count by the sample area (width × distance for transects, π × radius2 for point counts). At each replicate count, species locations are shuffled, and the outcome of their behaviours is different, ensuring not only stochasticity in model outputs, but also the independence of replicates.

Parameterisation of behavioural traits

To understand the effect of sampling method parameters on inaccuracy, four generic types of fish were created, representing behavioural traits that are known to affect accuracy on visual census methods, namely a “schooling” type, a “cryptic” type, a fish that is attracted to divers and a fish that evades divers (“bold” and “shy” types, respectively). For consistency we will use the same fish types from Pais & Cabral (2017), where a more detailed description can be found, along with videos and a detailed analysis of the isolated effect of behaviour on bias and precision.

Because urge vectors are an abstraction, and not something that can be directly measured, parameterisation must be pattern-oriented (Grimm & Railsback, 2012), based on the observable behavioural patterns that emerge from a combination of parameters. This is made easy by the NetLogo interface tools used to build FishCensus, where urge vector weights can be altered while the model is running. All four fish types were parameterised based on real species or families that are familiar to the authors, so that observation experience could aid in model parameterisation. Perception angle for cryptic fish was assumed to be 360°, due to the position of the eyes on top of the head and the predominantly sedentary behaviour. For all other fish types, a value of 320° was adopted, since it encompasses both visual and lateral line perception, in accordance with observations by Partridge & Pitcher (1980). ID distance, approach distance, schooling distance and some behaviour frequencies and patterns were parameterised by qualitatively matching behavioural patterns based on the authors’ experience from more than 250 UVC dives in temperate reefs, complemented by underwater video.

The schooling type is based on sparids from the genus Diplodus. These species usually form small schools and can be found shoaling on rock patches (Gonçalves et al., 2014). The cryptic type is based on blenniids from the genus Parablennius. These are small benthic fish that hide in crevices and males can have territorial behaviour in the reproductive season. Behaviours and frequencies were based on a study by Almada, Garcia & Santos (1987) and detectability values were based on MacNeil et al. (2008b). Both the shy and bold types share parameters from labrids from the genus Labrus. These species are solitary and reaction to divers varies with species and life stage.

In the absence of available data on fine-scale behavioural states and frequencies for the schooling, bold and shy types, these had to be roughly estimated from field experience which, in this case, is not a big concern if the general characteristics of the behavioural traits are present (e.g., Diplodus spp. are found stationary or shoaling more frequently than Labrus spp.).

The maximum cruise and burst speed values were calculated using the equations from Sambilay Jr (1990) from the caudal fin aspect ratio of representative fish species extracted from FishBase online database (Froese & Pauly, 2017). These were Parablennius pilicornis (Cuvier, 1829) for the cryptic type, Diplodus vulgaris (Geoffroy Saint-Hilaire, 1817) for the schooling type and Labrus bergylta Ascanius, 1767 for the shy and bold types.

Fixed attribute values for these types are specified in Table 1 and behavioural states and parameters for each type are summarised in Table 2. Fish types are available as online supplements in csv format and can be directly used as input files for the model (Datas S1–S4).

Table 1 Fixed attributes for the four types of fish used in the experiments.

See text for details.

 	Schooling	Cryptic	Shy	Bold	
Size (m)	0.2	0.1	0.3	0.3	
ID distance (m)	4	1	6	6	
Approach distance (m)	1.0	0.7	3.0	3.0	
Perception distance (m)	0.35	–	–	–	
Perception angle (degrees)	320	360	320	320	
Max. acceleration (m/s2)	0.2	0.1	0.1	0.1	
Max. sustained speed (m/s)	0.5	0.3	0.4	0.4	
Burst speed (m/s)	2.6	1.1	2.2	2.2	

Table 2 Behavioural states, frequencies and attributes for the four fish types used in the experiments.

(See text for details.)

		Schooling	Cryptic	Shy	Bold	
	Behavioural state	Wandering	Feeding	Stationary	Guarding	Feeding	Nested	Patrolling	Wandering	Stationary	Wandering	Stationary	
	Frequency	0.5	0.2	0.3	0.25	0.2	0.1	0.45	0.6	0.4	0.6	0.4	
	Detectability	1	1	1	0.3	0.6	0.1	0.5	1	1	1	1	
	Schooling?	TRUE	TRUE	TRUE	FALSE	FALSE	FALSE	FALSE	FALSE	FALSE	FALSE	FALSE	
	Schooling distance (BL)	1	1	1	–	–	–	–	–	–	–	–	
	Patch distance (m)	–	1	–	0.5	3	0.5	2	–	–	–	–	
Urge weights	Align	5	1	5	–	–	–	– 	–	–	–	–	
Centre	6	2	6	–	–	–	–	–	–	–	–	
Spacing	15	5	15	–	–	–	–	–	–	–	–	
Wander	3	1	1	3	3	0	3	7	7	7	7	
Rest	0	1	7	2	1	15	2	0	6	0	6	
Cruise	0	0	0	0	0	0	0	10	0	10	0	
Patch gathering	0	10	0	6	6	15	6	0	0	0	0	
Diver avoidance	10	10	10	4	10	0	10	10	10	−1	0	
Notes.

BL body lengths

Experiment

To understand the direction and strength of the effect of sampling method parameters on bias and precision, five values were picked for each of the 3 parameters in transects and five values for time and rotation speed for stationary point counts, while radius only varied in integers between 2 and 5, since a radius of 1 metre would be dominated by the effect of diver presence, and a radius of 6 would match the visibility limit, which is unrealistic for field studies (Table 3).

Table 3 Range of parameter values used to test methodology effects on bias and precision.

Transect	Values	Stationary	Values	
Length (m)	10, 20, 30, 40, 50	Radius (m)	2, 3, 4, 5	
Width (m)	1, 2, 3, 4, 5	Time (min.)	3, 5, 7, 9, 11	
Swim speed (m/min.)	2, 4, 6, 8, 10	Turning angle (º/s)	2, 4, 6, 8, 10	

Given that true density is unknown, Pais & Cabral (2017) used different densities based on the orders of magnitude found on previous field studies (e.g., Pais et al., 2014) , concluding that 0.3 fish/m2 ensured satisfactory computing performance while avoiding low baseline precision due to rarity. Therefore, true density (Dt) was fixed at 0.3 fish/m2 and 10 replicate runs were made for each combination of parameters to calculate the average estimated density (De) from the simulated diver counts. Bias was calculated as the absolute difference from true density and expressed as a proportion of true density: δ=|De−Dt|∕Dt.

Precision was calculated as the coefficient of variation of the estimated density from the 10 replicate counts and expressed as a percentage of true density.

The combined effects of all parameters on bias and precision were represented graphically using heatmaps. The significance of effects on bias was tested using multiple linear regression models including all main effects (three parameters per method) and interactions, after appropriate transformations to achieve linearity, normality of residuals and homoscedasticity on the full model. Mild deviations from assumptions were tolerated. Residual variance was extracted from the 10 replicate counts, and therefore the effects on precision were not tested for significance, as a single coefficient of variation results from all replicates. All analyses were made using R version 3.4.3 and the core stats package.

Results

For this study we addressed bias as an absolute deviation from true density, however, data and summary figures that maintain negative bias values are available as supplemental files (Data S5, Fig. S1), where it is shown that most of the values were positive (overestimations), particularly for mobile fish.

Stationary point counts had a larger bias and less precision on average across all behavioural traits and parameters. For both methods, the bold trait led to larger bias and cryptic behaviour corresponded to the smallest bias values. Schooling behaviour led to low precision in estimates for both methods, although the bold trait had the highest average coefficient of variation in transects (Table 4).

Table 4 Average and range of bias and precision values per behavioural trait across all sampling parameter combinations.

Bias and coefficient of variation (CV) are in percentage of true density (0.3 fish/ m2).

	Stationary	Transect	
	Bias (%)	CV (%)	Bias (%)	CV (%)	
Schooling	1,182.3	317.7	215.2	71.3	
(130.8–2,892.1)	(41.8–1,014.1)	(41.0–871.3)	(14.7–244.6)	
Cryptic	83.0	62.5	49.2	14.2	
(25.6–307.3)	(9.0–370.1)	(14.3–79.7)	(4.0–67.2)	
Shy	758	89.7	387.3	52.6	
(23.6–2170.2)	(33.5–202.9)	(32.7–1,730.0)	(14.7–198.2)	
Bold	2,940.2	282.5	857.7	90.3	
(478.5–9,181.4)	(60.0–1,082.8)	(102.7–4,200.7)	(17.5–305.1)	

Figure 1 shows heatmaps of observed (not modelled) bias and precision for every combination of method parameters on transects and stationary point counts. Regression models had a good fit in general (multiple R2 > 0.6), except for the cryptic trait where R2 was approximately 0.4 for transects and 0.2 for point counts. However, since the effects observed in Fig. 1 for cryptic fish are generally monotonic, the significance tests are analysed. For detailed results of the linear regression models see Table S1, and for raw response curves of bias and precision see Fig. S1. On stationary points (Fig. 1A), increasing survey time significantly increased bias for all behavioural traits, while increasing observation radius attenuates this effect for schooling, cryptic and bold fish. Shy fish count bias was not significantly affected by radius. High rotation speeds seem to slightly increase bias when a small radius is used, but overall the effect was not significant for schooling, cryptic and bold fish. On the other hand, bias increased with rotation speed for shy fish and was aggravated by long survey times and a large radius. While the patterns observed for precision are less pronounced (Fig. 1C), it seems like many of the parameters that tend to increase bias will also lead to higher variability. A long survey time will in general lead to more imprecise estimates, particularly for a small radius. Increasing radius tends to reduce variability, except for shy fish. Rotation speed does not seem to significantly affect precision unless radius is small.

Figure 1 Heatmaps of observed bias (% difference from true density) and precision (coefficient of variation) for all combinations of parameters and behavioural traits.

Darker shades represent more bias and variation. Each square represents average bias extracted from 10 replicate runs and the coefficient of variation of the same 10 runs is used to represent precision. Shading is scaled to the range of each behavioural trait (Table 4).

On transect surveys (Fig. 1B), a faster swim speed reduces bias for schooling, shy and bold fish. Cryptic fish surveys benefit from slower swim speeds, particularly on narrower transects. On the other hand, a narrower transect generates more bias for schooling, shy and bold fish. Transect length does not seem to impact bias for schooling species, but it has a very slight but significant effect in increasing bias for bold and shy species, particularly with slow speeds and narrow transects. In the case of cryptic fish, the opposite pattern occurs, with length reducing bias for slow and narrow transects. It is, however, evident in Fig. 1D that narrower, shorter transects will generally lead to less precise estimates irrespective of behavioural traits. Nevertheless, fast swim speeds usually increase precision and can attenuate this effect, particularly on mobile species.

Discussion

This study has provided an insightful overview of the combined effects of different methodological approaches in UVC, and how they differ for typically problematic behavioural traits. In fact, this exercise seems to confirm that the system is more complex than what is expected from an encounter probability model of randomly distributed moving objects, such as the Gerritsen-Strickler model (Gerritsen & Strickler, 1977). This was also observed by Watson, Carlos & Samoilys (1995) with the Reefex model, even though they mostly focused on the effect of speed and direction of fish. The present study helps establish a link between sampling method and accuracy using more realistic behavioural traits, and it is clear from our results that accuracy and encounter probability are not necessarily surrogates. In fact, Gerritsen and Strickler demonstrated that a predator could either increase the search radius or the speed to increase encounter probability; however, encountering more fish does not always mean being more accurate, and this is where behaviour seems to have different effects. In case of a shy species, increasing speed seems to be beneficial, but increasing radius in point counts increases bias. With cryptic fish, a faster transect will lead to more bias, while a wider radius is beneficial in point counts. This ultimately means that a stationary point is not always a special case of a transect with zero speed.

Dealing with the effect of behavioural traits is complex because they tend to affect different components of bias in UVC. The choice of traits for this study covers probably the most problematic cases for underwater surveys: fast movement can generate bias due to non-instantaneous sampling (Watson, Carlos & Samoilys, 1995; Ward-Paige, Flemming & Lotze, 2010; Pierucci & Cózar, 2015), cryptic behaviours or mimicry can lead to biased counts due to low detectability (Willis, 2001; MacNeil et al., 2008a; MacNeil et al., 2008b), and shyness or boldness can lead to bias due to observer influence (Kulbicki, 1998; Dickens et al., 2011). All these sources combined are very difficult to tackle and correct, and field methods usually tackle one or two components, most of the times having to compare estimated abundances with estimated “true” abundances (Sale & Sharp, 1983; St. John, Russ & Gladstone, 1990; Willis, Millar & Babcock, 2000; Bennett et al., 2009).

The fact that bias was mostly positive seems to go against the common assumption that UVC methods always underestimate (Willis, 2001; Colvocoresses & Acosta, 2007; Minte-Vera, De Moura & Francini-Filho, 2008). In fact, it seems obvious to assume that if some fish avoid divers and others are hidden or missed, we must be underestimating counts. However, since fish are coming into the sample area that were not there at the start, it is easy to get overestimations simply because of the non-instantaneous nature of UVC methods. This has been repeatedly shown by simulations and it is known to be linked not only to the direction of fish crossing the sample area (Watson, Carlos & Samoilys, 1995), but mostly to fish speed relative to the observer (Ward-Paige, Flemming & Lotze, 2010; Pierucci & Cózar, 2015; Pais & Cabral, 2017). In fact, Ward-Paige, Flemming & Lotze (2010) estimated positive biases of 672 to 1,100% in transect surveys for fish swimming at 0.4 to 0.6 m/s with an observer speed of 1 m/min., which falls within the orders of magnitude found in this study. This is the reason why we found that a faster speed in transects reduces bias for all mobile fish, by shifting observer speed towards fish speed. In the case of nearly-stationary cryptic fish, however, a fast swim will simply reduce the probability of detection and result in a slight underestimation (De Girolamo & Mazzoldi, 2001). This confirms the field observations by Lincoln Smith (1988), who seems to have correctly hypothesised that higher values due to fish speed could be overestimations. For this reason, it is very important to accurately parameterise fish speed on any simulation approach. In the case of FishCensus, fish speeds are a highly sensitive parameter (Pais & Cabral, 2017), and therefore they should be taken from real measurements, or at least be calculated from the caudal fin aspect ratio of the species using the equations from Sambilay Jr (1990), which can be done automatically in the model.

For the tested traits and across all combinations of sampling parameters, point counts had larger bias and more variability in estimates than transects. This confirms the observation made with FishCensus by Pais & Cabral (2017) using fixed dimensions with equal visibility and sample area for the same traits, but apparently contradicts the observations by Watson & Quinn II (1997) with the Reefex model, where point counts tended to have lower bias than transects with moving fish. However, the method simulated on the Reefex model was a top-down observation of a circle (instead of rotation in a cylinder), and fish were simulated with head-on movement, which is known to significantly increase bias on transects (Watson, Carlos & Samoilys, 1995).

In field studies comparing both methods, the general pattern is that point counts lead to higher estimated densities. However, different interpretations exist for this observation, from considering that higher is more accurate (Colvocoresses & Acosta, 2007) to recognising the possibility of overestimation (Kulbicki et al., 2010). Minte-Vera, De Moura & Francini-Filho (2008) found no differences with the shape of the sampling unit, but opted for point counts due to higher cost-effectiveness, also assuming that higher densities were better. In some cases, it is difficult to find differences between methods simply because of high residual variability that results in low statistical power (Samoilys & Carlos, 2000). In our simulations, a sample size of 10 was adopted as a compromise between feasibility in a field context and attained precision, an important aspect to consider given the cost of each sampling unit.

The observation that bold traits lead to higher bias is generally understood in the literature and is mainly due to an artificial gathering of a higher density of fish near the observer that leads to overestimation. This is known to contribute to overvalued reserve effects where the same species is more approachable and curious inside marine protected areas (Kulbicki, 1998; Willis, Millar & Babcock, 2000). An interesting pattern, which confirms previous simulations by Pais & Cabral (2017) and Ward-Paige, Flemming & Lotze (2010), is that stationary territorial fish, even with probabilities of detection as low as 10%, led to the lowest amount of bias when compared to mobile species, since they did not suffer from the effect of non-instantaneity. However, most field studies attempt to quantify UVC bias for cryptic fish, and one of the main reasons is that it is achievable. Their site-attachment and approachability make them ideal to estimate bias in the field, be it through poisoning (Willis, 2001), baited census (Stewart & Beukers, 2000), or by using a known number of golf balls as a proxy (Sayer & Poonian, 2007). These studies are very useful, but are focused on a single component of bias.

With a fixed sample size, increasing the area of the sampling unit will generally increase precision (Minte-Vera, De Moura & Francini-Filho, 2008), which was also confirmed by our results. However, precision is also significantly affected by behavioural traits (Pais & Cabral, 2017). Across all methodologies, schooling behaviour led to the lowest precision, as the difference between counting or missing a school is usually of at least tens of individuals. In transects, however, swimming across a larger area with a chance of attracting bold fish along the way leads to a greater effect of this trait in reducing precision, as more encounters will lead to much higher counts (Colvocoresses & Acosta, 2007). This is particularly relevant when establishing sample sizes for monitoring programs. Pais et al. (2014) and Jones et al. (2015) observed how sites with flat or patchy habitat led to low precision in fish counts and thus required a higher sample size to achieve the same power to detect changes. The fact that schooling is a known strategy of many species occupying these habitats is another point in favour of being precautious with sample sizes if resources allow. This means conducting a priori power analysis with schooling and/or bold species in flat or patchy sites and then establishing the sample size for all sites based on this reference.

This study illustrated the difficulty of objectively establishing methodologies based on field data alone, as sample dimensions and observation times interacted with each other and with species behaviour. In point counts, increasing survey time increased not only bias but also variability. This results from having a stationary observer counting mobile fish that enter the sampling unit. As time passes, the estimated abundance will only tend to increase, eventually leading to overestimation (St. John, Russ & Gladstone, 1990; Watson & Quinn II, 1997). This is similar to the “first phase” and “edge effect” identified by Kulbicki et al. (2010) at the beginning and end of transects, where the surveyor stays on the same spot for a longer period and counts fish that gathered around during the setup phase, or new fish that keep coming into view and being counted when the diver has already reached the end of the transect. This effect is not incorporated into the current version of the FishCensus model, since the diver starts and finishes the transect immediately, but it would certainly result in larger overestimations of mobile fish with transects, particularly with bold traits.

Bias in point counts was reduced for most species by using a larger radius, and the same was observed when increasing width in transects. In fact, several field studies have observed that density estimates are lower when the sample unit area increases (Sale & Sharp, 1983; Colvocoresses & Acosta, 2007; Minte-Vera, De Moura & Francini-Filho, 2008; Prato et al., 2017). This tends to occur not only because there may be a reduction of detectability with distance for some species (Bozec et al., 2011; Katsanevakis et al., 2012), but also because the increase in area may not be proportionately met by an increase in fish numbers within that area, and a slightly higher count is divided by a much larger area (Colvocoresses & Acosta, 2007). Our results show that lower density estimates reduce bias for most species, since smaller units were already overestimating counts. However, this is not the case for cryptic fish in transects, where counting less fish results in an even larger underestimation, thus increasing absolute bias (Lincoln Smith, 1989; Willis, 2001). For shy fish in point counts, the pattern is inverted. Since the diver is stationary and shy fish are keeping their distance (approximately 3 m in our case), increasing the radius will result in a higher count, which increases overestimation as the diver rotates and more fish enter the area.

In strip transects, besides the effect of observer speed, which we already discussed, transect dimensions can also affect bias and precision. In general, the effect of transect length on bias was not very strong, except in the case of slow swimming surveyors. In this case, a long transect increased bias of mobile species by worsening the effects of non-instantaneity (Lincoln Smith, 1988; St. John, Russ & Gladstone, 1990; De Girolamo & Mazzoldi, 2001). On the other hand, cryptic fish counts were slightly more accurate with long, slow transects, as the probability of a fish becoming visible while the observer is still looking increases. Very short transects, on the other hand, led to low precision. This is the result of a sample unit area that is too small to capture the heterogeneity of fish distribution (Minte-Vera, De Moura & Francini-Filho, 2008).

It is widely accepted that there is no universal method for all possible conditions and species. However, even authors applying in-depth optimisation approaches with empirical data recognise limitations due to the complexity of the system (Jones et al., 2015). In the end, a single value for the estimated density of a species is the result of the relative position of several fishes and one surveyor in space and time, bounded by a combination of parameters that define a sample unit. Where other analytical or field methods struggle to capture this complexity, agent-based spatial simulations thrive, making them potential candidates as decision-support tools.

Simulations, however, are not a replacement of the real system, and this approach should be complementary to field experiments. This does not necessarily mean fitting or validating the model with field data, given that it was created because of a real value that is unknown, but rather using the model to help interpret real observed patterns, as we did repeatedly in this discussion. While modelling facilitates understanding by simplifying the system, it requires certain assumptions. In the case of FishCensus, the lack of individual variability in terms of size and behavioural repertoire, absence of habitat complexity, two-dimensional representation and a rather unforgiving implementation of diver memory can be pointed out as the main simplifying assumptions (Pais & Cabral, 2017).

It should also be noted that we used a single species approach. While single species assessments are common in fisheries management surveys (Gardner & Struthers, 2013), often whole community assessments are important, which include even more sources of bias (Lincoln Smith, 1989; De Girolamo & Mazzoldi, 2001; Babcock, Egli & Attwood, 2012; Henriques et al., 2013a; Henriques et al., 2013b; Pais et al., 2014). Besides the surveyor having to focus on a larger number of fish, which will likely favour the most conspicuous (Willis, 2001), there is also species, age and gender-specific distribution and behavioural traits (MacNeil et al., 2008a; MacNeil et al., 2008b; Kulbicki et al., 2010). The FishCensus model supports multiple “fish types” simultaneously, but a single species assessment was deemed more adequate and advantageous, not only because this reduces computing time, but also because it isolates the particularities of each behavioural trait, so they can then be sampled together if similarly affected by the choice of method.

Conclusions

To minimise bias in underwater visual surveys, our results suggest that a relatively large radius and short time should be favoured in point counts, keeping in mind that rotation speed will not have a marked effect in these conditions. A large radius will increase bias for shy fish, but if survey time is short, it should not be significant. In the case of strip transects, swimming fast along a wide transect favours smaller bias for mobile species. For cryptic species, a slow swim along a narrow transect should be favoured. It should be noted that very narrow and very short transects will lead to less precision due to the small size of the sampling unit.

Our results support an objective approach to sampling design in UVC, not only in terms of the amount of replication required (Pais et al., 2014; Jones et al., 2015), but also starting on the sampling methodology itself. Abandoning traditional methods can disrupt long time series, but probably in most cases we can afford a change that will benefit future surveys (Caldwell et al., 2016). The recognition of this problem is evident as new approaches are proposed to address the shortcomings of established methods (Minte-Vera, De Moura & Francini-Filho, 2008; Kruschel & Schultz, 2010; Prato et al., 2017). Using at least two different methods for multispecies assessments (Lincoln Smith, 1989; De Girolamo & Mazzoldi, 2001) is supported by our findings. In this case a survey should start with a faster scan of a larger area for mobile species, and then move on to a more focused, slower approach on a smaller area for sedentary, smaller species. This makes the reasonable assumption that the diver presence on the first observation period does not significantly affect subsequent counts of site-attached species (Sayer & Poonian, 2007).

It must be stressed that the results obtained apply to the particularities of the species that inspired each behavioural trait. Ideally, one should attempt to replicate the behaviour of the exact species of interest as realistically as possible within the model, and then use it to optimise, plan and interpret field surveys.

Supplemental Information

Article S1 Full ODD Model Description

Detailed description of the model according the the ODD protocol (Overview, Design concepts, Details) for describing agent-based models.

Click here for additional data file.

Data S1 Input file with the schooling fish parameters

This file can be used directly as input in the FishCensus model.

Click here for additional data file.

Data S2 Input file with the cryptic fish parameters

This file can be used directly as input in the FishCensus model.

Click here for additional data file.

Data S3 Input file with the shy fish parameters

This file can be used directly as input in the FishCensus model.

Click here for additional data file.

Data S4 Input file with the bold fish parameters

This file can be used directly as input in the FishCensus model.

Click here for additional data file.

Data S5 Raw data

Raw data for all the model runs, including bias calculated for every replicate.

Click here for additional data file.

Table S1 Multiple linear regression models

Detailed summaries for the multiple linear regression models for all fish types. (A) Summary tables for point counts. (B) Summary tables for transects.

Click here for additional data file.

Figure S1 Raw response curves of bias to different methodologies

Average bias calculated from 10 replicates as the difference between estimated and true abundance, divided by true abundance. Positive bias corresponds to an overestimation of fish density. Precision calculated as the standard deviation of the 10 estimates, divided by true density. A radius of 1m for point counts was not used in the analyses but its effect is shown. (A) Bias in point counts. (B) Precision in point counts. (C) Bias in transects. (D) Precision in transects.

Click here for additional data file.

The authors acknowledge everyone who tested the model and interface, Christine Ward-Paige for suggestions and discussions regarding the AnimDens model, Uri Wilenksy for NetLogo and code snippets for vector-based swarming, Kenneth A. Rose for valuable feedback and suggestions on an early version of the model and João G. Rosa for revising the calculation of drag forces.

Additional Information and Declarations

Competing Interests

Author Contributions

Data Availability

The authors declare there are no competing interests.

Miguel Pessanha Pais conceived and designed the experiments, performed the experiments, analyzed the data, prepared figures and/or tables, authored or reviewed drafts of the paper, approved the final draft.

Henrique N. Cabral conceived and designed the experiments, analyzed the data, approved the final draft.

The following information was supplied regarding data availability:

FishCensus model code and documentation is available at the CoMSES OpenABM database: https://www.comses.net/codebases/5305/

Version 2.0 of the model was used in this paper.

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
