# Peer review of "Effect of underwater visual survey methodology on bias and precision of fish counts: a simulation approach"

_PeerJ, doi:10.7717/peerj.5378_

## Round 0.1 · original submission · Major Revisions

The methodology and the contents of the manuscript are essentially good but I believe that following the reviewers suggestions will make this a better paper. They aren't really major revisions but require some further work. I hope that the authors consider resubmitting the improved version.

Reviewer 1 ·

Basic reporting

The paper will be a very good contribution on the topic once some of the roughness in model description, interpretation of results, and general language is dealt with. I found the description of model parameters to be extremely lacking and it took several reads to sort things out. The first three paragraphs of discussion do not touch on your results in any sense. I found this odd as it should be the showcase. Not some long discussion that really reads like an introduction. I want to know what it is you found that should captivate me as a reader, and then get into the literature supporting it. Structurally I was searching for the point until I hit paragraph four when you actually start talking about what your model performance.

In my mind this is a Gerritsen-Strickler encounter probability model and the diver speed is just confounding within the area sampled issue. The way point samplers deal with this is by using habitat based designs and ramping up sample to collect increase information over area. I think this needs to be dealt with in paragraph one of your discussion. It’s the main point of your results and to me it’s pretty clear in your results. The explanation of the model results in the discussion really needs to be rethought and reorganized because it doesn’t flow as written and I struggled making sense of what you believe were the most important outcomes.

Experimental design

The fundamental design is fine. The execution of describing the model is lacking.

For instance there are few biological or mathematical descriptions of fish traits. What does diver avoidance = 10 actually mean? I have no idea, is this a function of some kind? I can’t decipher how this manifests in behavior in the model? What is patch gathering? What is centre?

You describe some pretty common ‘behaviors’ such as schooling or cryptic but you don’t actually tell the reader how that is modeled. For instance schools really act as a single individual in space, but from what I could tell you’re modeling individuals to simulate a density of 0.3 per meter. Are you simply distributing the school spatially? Do they aggregate in the model? Schools by definition are extremely patchy because all the individuals occur in one sighting. At one point you state that abundance apparently does not affect bias. Yet when I think about counting a school of a thousand fish as a diver, with no permanent record, all I can think of is bias. Memory in the sense of what you modeled should dramatically reduce the ability of a diver to detect ‘individuals’ in a school and bias these counts severely. Yet I don’t understand how that’s handled because don’t know how individuals in schools move relative to one another, how they’re actually spatially distributed on your grid, and how that might manifest itself in ‘counting and density errors’ in your model.

Units are difficult to sort out relative to time-step, fish movement speed, cell size, and count-memory. I have no idea how the perception distance, perception angles etc are actually used in the model to understand how those may impact model results (i.e. fish response to vehicle is not really described other than to say that you have some parameters). This all needs better description, not just redirection to another paper or effort somewhere else. It’s not the readers job to hunt down supporting arguments if those can’t be easily explained within the context of a stand-alone effort.

Validity of the findings

I don’t really understand the estimation of density on the point count census. I understand it from a transect point of view because its distance. However you state the for the point count, the thing is rotating, but do not describe if the swept area of that cube is additive or static? More importantly, in either sampling case, time is confounded with area and speed. You don’t really address this in a straight forward way. Think about the stationary point count, as an extremely slow vehicle in which area basically doesn’t change and yet survey time is increased incrementally. Thus encounter probability in the volume is increasing, yet area/volume is static. Nobody in their right mind should be deriving densities in this manner from empirical data, specifically for highly mobile species. You’ve got that correct in here. Now think about the moving diver case. When you increase speed and decrease distance all you’ve really done is decrease encounter probability. Vice versa, decrease speed and increase transect length, you’ve increased encounter probability. Hence time is confounded within the transect information. Whereas you explicitly deal with it on the point count. In my mind this is a Gerritsen-Strickler encounter probability model and the diver speed issue is confounded with the area issue. The way point samplers deal with this is by using habitat based designs and ramping up sample (i.e. get data over more area). I think this needs to be dealt with in paragraph one of your discussion. It’s the main point of your results and its pretty clear in your results (heat plots).

Line 253. I think this is critical and needs to be in the results or somehow highlighted. Don’t think it’s enough to just call it bias. I like the heat plots, but maybe you can incorporate color to determine positive/negative bias results? Or 3 dimensional.

Line 254-255. Maybe that’s the case in the literature but I would not have assumed that these were overestimates at all, particularly in the case of mobile schooling species with strong ‘boldness’ factors.

Line 285-287. No idea what this actually means.

Line 301. You approach the detection probability relative to movement of the sampler/diver (i.e. non-instantaneous sampling) but don’t put it into a single package. A stationary point count, is just a really slowly executed transect. The more you increase time, the higher the encounter probability gets and the worse the estimate. Transects are just confounded with speed and distance.

Line 307-308. Fits with just what I said, but in this case you increase the area and thus relieved some of the assumptions about encounter probability all of which is confounded by the fish movement.

All of this doesn’t really deal with attraction and avoidance, where there is active ‘seeking’ behavior where divers are trailed or avoided. It’s sort of dealt with as shy or bold in approach distance, but that doesn’t really describe the behavior of fishes that follow (e.g. barracudas).

Additional comments

I definitely believe this is an effective model and deserving of publication but not in it's current state. Model description needs to be sorted out so that the interactions dealt with above can be sorted out by the reader. Otherwise it's sort of a mixed bag of things that take a lot of effort by the reader to understand the point.

Reviewer 2 ·

Basic reporting

The introduction was very clear and laid out the problem with designing surveys for reef fish very well. The literature appeared relevant. The structure of the paper followed normal patterns. I may have missed it, but there didn’t appear to be any raw data provided? The model parameters for the simulations are provided. The tables were informative and followed from the text. A couple of comments on the figure.
1) The manuscript would be improved by showing some of the relationships in graphical form that are observed in the S1 and S2 tables. Patterns in relationships among variables using the raw simulation results would be nice to see here. I suggest adding some multi-panel figures that show some of the interesting results from the bias analyses, at least as additional supplementary material.
2) The lone figure is difficult to interpret. I would recommend splitting this figure by species type and showing the results for each type in a single figure (with CV and bias). This would greatly simplify the information and make it easier to follow the patterns being described in the results.
3) Could the authors present an additional table with the recommended (based on the simulation) sample design and the most attainable values for bias and CV? The current Table 4 is interesting in that the values for bias and precision on average are quite high when combined and likely none of them (on average) would be a sutiable level of bias or CV. It would be more informative to present the best version of survey design and the likely bias and CV.

Experimental design

The research seems appropriate for PeerJ. The research question is well defined and survey design for underwater visual surveys is has a number of knowledge gaps for which this simulation provides a nice tool to examine. The experimental design seems appropriate for the simulations, the authors do a nice job of presenting information in to back their choice of species groups/parameter combinations. Two main comments here
1) It is not quite clear where the choice of density of fish was derived from. Also, I couldn’t tell if this was the overall density of fishes across all 4 groups or whether this was the density of each fish group.
2) There could be few more details provided on the mechanics of the simulation model. For example, the reader has to refer to the original contribution to understand where the stochasticity in the simulation came from.

Validity of the findings

The simulations presented in this manuscript appear to be robust and represent the natural variability in conditions the authors have observed in reef fish. As the authors recognize in the discussion, the specific results of this simulation are relevant to the characteristics of the fishes they have specified parameters for. The value of this paper is to demonstrate how the simulation model can be used to explore other types of fishes with other characteristics. As such it is novel, but the more novel contribution would be in the original paper that set out the simulation (i.e. Pais and Cabral 2017). It also seems that a fair amount of work has been done previously on these questions, but the previous results are well integrated into this manauscript. The conclusions that the authors reach appear logical given the data and simulations presented and the data seem robust and statistically sound.

Additional comments

I enjoyed reading this paper, it was an interesting look at a simulation model for an important survey question and the methods here could be applied elsewhere.
1) The abstract is a bit confusing to the reader that has not read the paper. It was not initially clear to me what “bold” and “sly” fish were or what a “slow swim” referred to. Probably because I was not aware the authors were talking about diver surveys until reading the introduction (as there are other types of underwater visual surveys). I would suggest defining this in the abstract, as well as some of the other terminology that appears in the abstract, such as by saying “bold fish (fish attracted to divers)”.
2) It was interesting that a single density was chosen for all the simulations. It would seem from other studies of this type that the true density has a large effect on CV and potentially bias (for rarer species). Also, some of these species groups are likely to have very different underlying densities on reefs. For example, pelagic schoolers might be an order of magnitude more dense than cryptic species. I would be interested to hear a bit more about justification of this number across the different species groups.

---

## Round 0.2 · accepted · Accept

As mentioned in the previous review round the changes where not really major. I'm glad your changes were able to correct the problems reported by the reviewers. I apologize to the authors for any delays in processing this paper caused by my unavailability.

# Reviewer 2 ·

Basic reporting

The revisions and supplemental material have addressed my primary concerns about the basic reporting. The figure is still difficult to interpret.

Experimental design

The supplemental text in the manuscript and supplemental material has improved the description of the experimental design.

Validity of the findings

The improvements to the manuscript have addressed any concerns about the validity of the findings.